

# Analysis of epidemic trend of respiratory pathogens in children after long-term pathogen isolation

Ping Chen[1], Yahong Li[2], Shiwei Li[2,3], Xin Hua[2] and Yu Liu[2]

[1] Geriatrics Department, Tianshui First People's Hospital, Tianshui, Gansu, China
[2] Laboratory Department, Tianshui First People's Hospital, Tianshui, Gansu, China
[3] Department of Basic Medicine, Xi'an Jiaotong University, Xi'an, Shaanxi, China

Corresponding author
Shiwei Li, 463418404@qq.com

## ABSTRACT

**Background.** China implemented strict prevention and control measures during the COVID-19 pandemic, leading to prolonged home confinement of children and significantly reduced exposure to respiratory pathogens. The ability of children under these conditions to resist respiratory pathogens post-lifting of containment measures remained unknown. The children's ability to face respiratory pathogens post-isolation was assessed through collecting respiratory pathogens detection data, statistical analysis and comparing with pre-epidemic data. In this way, we addressed data gaps in related fields and provided empirical support for research and public health sectors.
**Methods.** The pathogen detection method was real-time polymerase chain reaction (PCR). Data analysis software: SPSS Statistics 23.
**Results.** Four environmental factors—temperature, humidity, wind speed and sunshine time—exerted interactive effects ton respiratory pathogens prevalence. Influenza A and B viruses, *Mycoplasma pneumoniae* and adenovirus exhibited seasonally prevalent patterns in winter, while respiratory syncytial virus showed lower prevalence in summer, and rhinovirus had no significant seasonal variation. The infection rate of respiratory syncytial virus and the number of expected infections decreased with the increase of age, whereas adenovirus, influenza virus, *Mycoplasma pneumoniae* and bacterial pathogens showed age-related increases in both metrics. Gender-related differences were observed in higher infection rates of *Klebsiella pneumoniae*, *Haemophilus influenzae* and *Pseudomonas aeruginosa* in men compared to women. Co-infection analysis revealed that respiratory syncytial virus exhibited the highest co-infection diversity, followed by influenza B virus, *Mycoplasma pneumoniae*, and *Staphylococcus aureus*. *Pseudomonas aeruginosa* showed inhibition of co-infection with *Klebsiella pneumoniae*. *Streptococcus pneumoniae* had been shown to inhibit co-infection with *Haemophilus influenzae*, *Staphylococcus aureus*, and *Klebsiella pneumoniae*. The lower respiratory tract infection rate of *Mycoplasma pneumoniae* and bacterial pathogens exceeded the overall infection rate. Infection rates of influenza B virus and adenovirus were lower than the overall rate, while those of influenza A and rhinovirus paralleled the overall rate. Analysis of statistical data from China and Gansu Provincial Centers for Disease Control and Prevention revealed that most respiratory pathogens showed an increase in the infection rates post-COVID-19, particularly respiratory syncytial virus, indicating altered prevalence patterns in children due to prolonged exposure to isolated pathogens.

## INTRODUCTION

Acute respiratory infections (ARIs) impose a substantial global health burden, ranked by World Health Organization (WHO) as the fourth leading cause of mortality worldwide, with marked socioeconomic and clinical ramifications (*Vos et al., 2020*). Child health is severely impacted by ARI, which has become the leading cause of hospitalizations and mortality among children, particularly in low- and middle-income countries (*Shi et al., 2017*).

During the pandemic, countries adopted diverse prevention and control measures. China had adopted strict containment measures. During this period, population movement was significantly restricted, and the infection rate of other respiratory pathogens among children had been greatly reduced while controlling the COVID-19. COVID-19 prevention and control measures had been gradually relaxed after 2022. The effects of prolonged absence of respiratory pathogens exposure on immune system development and infection response in children remained unclear. This posed a huge challenge to child health and public health. Few large-scale studies exist in related fields. This study was established in the context of prolonged home confinement during the COVID-19 pandemic, which reduced children's exposure to respiratory pathogens, leading to the development of a large-scale model of long-term pathogen isolation. This study aimed to fill the knowledge gap regarding the effects of long-term pathogen isolation on children's respiratory pathogen responses by comparing prevalence data before and after the COVID-19 epidemic, considering factors like environment, gender, and age. In order to provide specific cases for relevant research and public health institutions.

## MATERIALS & METHODS

### Study population

The study cohort comprised 8,720 pediatric patients (aged 0–13 years) receiving clinical care in the First People's Hospital of Tianshui during the period from April 2023 to March 2024 (Data S1). A total of 9,670 respiratory specimens were collected, with 7,670 allocated to viral detection (including *Mycoplasma pneumoniae* categorization) and 2,000 to bacterial nucleic acid analysis. Study data were derived predominantly from hospitalized individuals with clinically significant respiratory infections, excluding those with mild or latent presentations. Clinical data were documented in the hospital information system by the pediatricians, encompassing gender, age, physical signs, diagnosis, and diagnostic test results. Specimens were collected and transported to the laboratory by certified nursing staff. Specimens underwent laboratory processing with confirmatory testing as clinically indicated. The laboratory maintained CNAS-accredited quality management systems with rigorous protocol adherence in quality control and evaluation. Written informed consent forms were obtained from all participants before enrollment: A complete description of the research purpose, process, potential risks and benefits, privacy measures (such

as data anonymization), the right to voluntary participation, and opt-out mechanisms at any time. For participants with limited educational backgrounds, a trained physician would provide a verbal explanation, followed by a relative witnessing and signing upon confirming comprehension. The Ethics Committee oversaw the study's implementation and conducted annual reviews of its ethical compliance.

## Definitions

1. Patients with ARIs must meet at least one of the inclusion criteria as follows:

    (1) Fever (axillary temperature ≥37.2);

    (2) White blood cell counts were normal or low, and the proportion of lymphocytes was elevated;

    (3) White blood cell counts and neutrophil ratio increased;

    (4) Cough, sore throat, wheeze, expectoration, chest pain, moist/dry rales;

2. Lower respiratory tract infections (LRTI) cases were defined by clinical diagnoses of bronchitis, of pneumonia (including lobar pneumonia), emphysema, pulmonary infections, respiratory distress syndrome, and respiratory failure.

3. In this study, co-infection refers to simultaneous infection of greater than or equal to two pathogens. Pathogens detected include: influenza virus A (IFV-A), influenza virus B (IFV-B), respiratory syncytial virus (RSV), human rhinovirus (HRV), human adenovirus (HAdV), *Mycoplasma pneumoniae* (MP), *Staphylococcus aureus* (SA), *Klebsiella pneumoniae* (KP), *Legionella pneumophila* (LP), *Haemophilus influenzae* (HI), *Pseudomonas aeruginosa* (PA), *Streptococcus pneumoniae* (SP).

## Statistics and ethics

The statistical software was SPSS Statistics 23 (IBM Corp., Armonk, NY, USA). Statistical analyses comprised binary logistic regression, Poisson regression and negative binomial regression. Select a regression model based on whether the mean was approximates the variance. Binary logistic regression used the Hosmer-Lemeshow test (HL) to determine the fit degree of the model, $p > 0.05$, as the inclusion criteria. Poisson regression (PR) and negative binomial regression (NBR) used omnibus test (OT) to determine whether the independent variable had a significant effect on the dependent variable, and $p < 0.05$ was used as the inclusion criterion.

This study was approved by the Ethics Review Committee of the First People's Hospital of Tianshui and all participants gave a signed informed consent. The IRB approval number was 2023-07.

Tianshui is located in the northern hemisphere, temperate zone, the terrain is a valley, the environmental data during this period from Tianshui Meteorological Bureau.

## Experimental method

In this study, the detection method was QRT-PCR, the detection reagent came from Shengxiang Biotechnology Co., LTD (Jinzhou, China). Reagent batch number: respiratory pathogen nucleic acid (IFV-A, IFV-B, RSV, HRV, HAdV, MP) 23026-2, respiratory pathogenic bacterial nucleic acid (SA, KP, LP, HI, PA, SP) 23002. China Food and Drug Administration (CFDA), State instrument registration number: 20213400256 and

**Table 1  Minimum detection limits for pathogens.**

| Pathogens | Detection limit |
|---|---|
| IFV-A | $2.0 \text{TCID}_{50}$/ml |
| IFV-B | $2.0 \text{TCID}_{50}$/ml |
| RSV | 500.0 copies/ml |
| HAdV | 500.0 copies/ml |
| HRV | 500.0 copies/ml |
| MP | 500.0 copies/ml |
| SA | 2,875 CFU/ml |
| KP | 900 CFU/ml |
| LP | 340 CFU/ml |
| HI | 625 CFU/ml |
| PA | 675 CFU/ml |
| SP | 15 CFU/ml |

20223400597. Nucleic acid extractor was NT-9600 of Shengxiang Biotechnology Co., LTD (Jinzhou, China). The amplification instrument was the MA-6000 from Yari Biotechnology Co., LTD. Patient samples 300 µl were detected. Reaction system included PCR reaction solution 43.5 µl, enzyme mixture 1.5 µl, patient sample 5 µl (after extraction). PCR reaction conditions included an reverse transcription at 50 °C for 30 min, initial denaturation at 95 °C for 1 min, followed by the 45 cycles of denaturation at 95 °C for 15 s, annealing, extension and fluorescence detection and at 60 °C for 30 s a final extension at 72 °C for 7 min. The minimum detection limits of pathogens in this study were shown in Table 1. All pathogen detection in this study was completed in the laboratory of the Laboratory of Tianshui First People's Hospital, which passed the CNAS laboratory quality management system certification, which is derived from ISO15189. The experimental instrument has been calibrated and its performance verified, the experimental method has been verified, the experimental reagent has been verified by the batch number, and the personnel have been trained and have a job certificate. The set temperature of the laboratory was 22 °C, the humidity was 35% RH, and the laboratory was disinfected twice a week. Each experiment contained 1 negative quality control and 1 weak positive quality control. The results of each experiment were evaluated, and specimens with poor amplification curves and other possible inaccuracies were re-examined. The CT value less than 38 was considered as the positive criterion.

# RESULTS

## Analysis of factors related to the number of pathogens infected

Table 2 summarizes meteorological parameters of the Tianshui region. The Tianshui region exhibits a temperate climate characterized by three-month meteorological seasons. with spring occurring from March to May. The data presented in this table comprehensively outlines the environmental characteristics of the Tianshui area for readers. Figure 1 demonstrates the epidemiological patterns of positive detection rates for various respiratory pathogens over time: IFV-A, IFV-B, MP and HAdV all have obvious one-month infection

**Table 2  Weather data of Tianshui.** In Tianshui, spring is from March to May, summer is from June to August, autumn is from September to November, and winter is from December to February of the following year. The data presented in this table aimed to comprehensively and clearly outline the environmental characteristics of Tianshui area for readers, and provided data for the data analysis of Table 3.

| Month and environmental factor | Mean wind speed (m/s) | Mean air temperature (°C) | Mean relative humidity (% RH) | Sunshine duration (h) |
|---|---|---|---|---|
| 2023.04 | 1.6 | 12.9 | 57.1 | 172.5 |
| 2023.05 | 1.6 | 16.9 | 62.7 | 151.5 |
| 2023.06 | 1.3 | 21.2 | 60.7 | 206 |
| 2023.07 | 1.4 | 23.9 | 63.8 | 197.1 |
| 2023.08 | 1.6 | 23.5 | 65.2 | 203.3 |
| 2023.09 | 1.3 | 19.2 | 73.3 | 116 |
| 2023.10 | 1.1 | 11.8 | 79.3 | 122.1 |
| 2023.11 | 1.3 | 5.9 | 61.3 | 157.7 |
| 2023.12 | 1.2 | −0.6 | 58.1 | 162.5 |
| 2024.01 | 1.3 | −1 | 55.8 | 136.2 |
| 2024.02 | 1.6 | 0.9 | 59.1 | 102.1 |
| 2024.03 | 1.3 | 8.8 | 56.4 | 177.3 |

peaks in winter; The detection rate of RSV remains relatively low in summer, while HRV exhibits no significant seasonal variation. The infection level of HRV remained relatively stable throughout the year. The typical high-incidence periods of bacterial pathogens are June, November and January. The distribution of infections exhibits marked seasonal variation. Table 3 included positive detection data, environmental factors, age, gender and other variables for statistical analysis, revealing that IFV-A and IFV-B infections peaked in winter, while KP infections were more prevalent in summer. The expected infection counts attributable to viral pathogens are the highest in the 1–6 age group, followed by the <1-year old age group, and the lowest in the >6-year age group. The expected infection counts for bacterial pathogens decline with age, whereas those for MP increase with age. In terms of gender differences, the expected number of men infected with KP, HI and PA was significantly higher than that of women.

## Pathogens and age

The data in Table 4 and Fig. 2 indicated that the male-to-female ratio among patients undergoing respiratory pathogen testing was 1.45:1 ($n = 5,729:3,941$), with similar positive ratios of various pathogens between males and females. With increasing age, the RSV infection rate declined from 52.35% to 6.06%, while HRV rates slightly decreased. In contrast, IFV-A, IFV-B, HAdV and MP rates gradually increased, whereas SA, KP and PA rates decreased progressively. The infection rate of COVID-19 among children in this area was extremely low, with a positive rate of 3.4% (total $n = 449$). The results indicated that HI and SP detection rates were relatively high across all patient groups. By comparing the procalcitonin (PCT) results of the patients, it was found that 12.3% of the patients positive for HI had elevated PCT, and 12.8% of the patients positive for SP had elevated PCT. Therefore, most HI and SP might be parasitic bacteria.

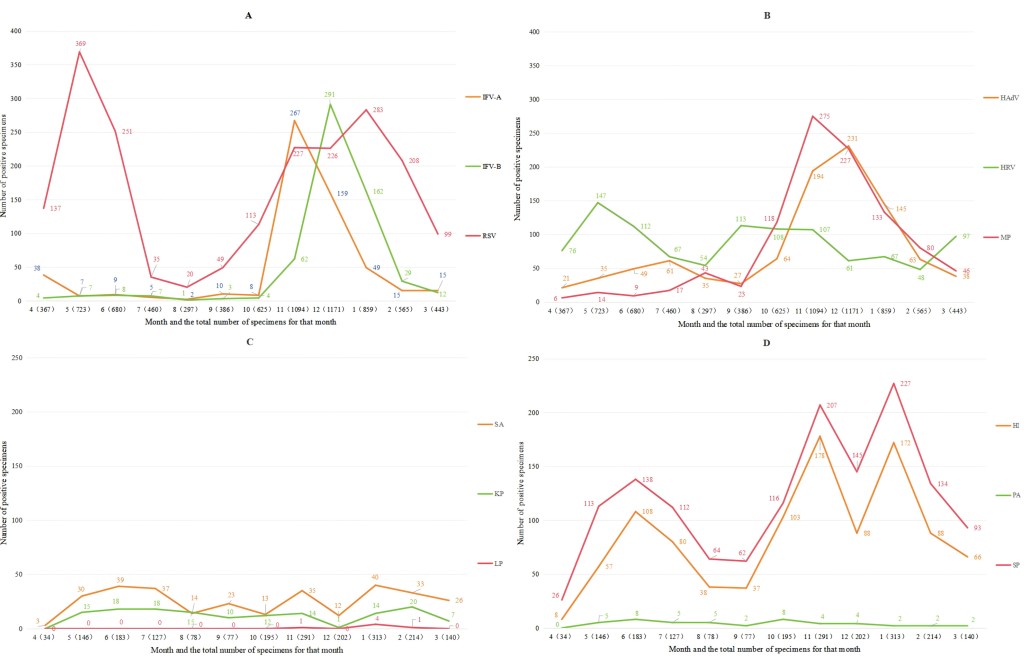

**Figure 1    The number of pathogen infections changed with time.**

## Co-infection

To investigate pathogen co-infection patterns, this study conducted a statistical analysis of 950 patients with 12 pathogens simultaneously detected, and performed pairwise examination of pathogens in the cases involving co-infections of more than two pathogens. The data were shown in Fig. 3. The results showed that the co-infection between HI and SP with other pathogens was the most prevalent, primarily due to the large number of their parasitic infections. In this study, to determine the relationship of co-infection among pathogenic organisms, these two pathogenic bacteria were excluded. Among the pathogens co-infected with IFV-A, MP accounted for the highest proportion at 18.96% (11/58). The IFV-B was MP 9.76% (4/11), RSV was SA 16.07% (45/280), HAdV was MP 13.64% (12/88), HRV was RSV 24.78% (28/113), MP was SA 8.45% (12/142), SA was RSV 30.41% (45/148), KP was RSV 27.63% (21/76), PA was KP 29.03% (9/31), HI was RSV 21.58% (112/519), and SP was RSV 27.94% (197/705). The co-infection patterns among 950 pathogen test results were analyzed using statistical software (Table 5). The data revealed that RSV demonstrated the strongest co-infection potential, followed by IFV-B, MP and SA. The co-infection risk between IFV-B and RSV was the highest, with an estimated coefficient of 20.651 (95% CI [2.797–152.468]). Notably, PA and KP exhibited the ability to inhibit mutual co-infection. SP could inhibit co-infection with HI, SA and KP. This table displayed the predicted data only when the pathogen was present in the respiratory tract. Its pathogenicity and the impact of its parasitic state on other pathogens could not be inferred from this table.

**Table 3 Statistical analysis of factors affecting the number of pathogen infections.** Including gender, age, season, mean wind speed, mean air temperature, mean relative humidity, sunshine duration in the table. Poisson regression and negative binomial regression were used for statistical analysis. N indicated non-significant, placed in the table for data completeness and comparison.

| Dependent variables and statistical models | Independent variable | Significance | Exp(B) | 95% Wald confidence interval for Exp(B) | |
| --- | --- | --- | --- | --- | --- |
| | | | | Lower limit | Upper limit |
| IFV-A | Spring | 0.020 | 10.175 | 1.443 | 71.768 |
| NBR | Winter | 0.008 | 233.337 | 4.214 | 12,919.363 |
| OT $p = 0.000$ | Autumn | 0 | 582.19 | 28.007 | 12,102.312 |
| | Summer | Contrast | 1 | . | . |
| | <1 | 0.76 (N) | 0.885 | 0.403 | 1.942 |
| | 1–6 | 0.001 | 3.385 | 1.656 | 6.920 |
| | >6 | Contrast | 1 | . | . |
| | Sunshine duration | 0 | 1.055 | 1.024 | 1.086 |
| | | | | | |
| IFV-B | Spring | 0.374 (N) | 2.637 | 0.31 | 22.410 |
| NBR | Winter | 0.050 | 80.956 | 1.006 | 6,512.531 |
| OT $p = 0.000$ | Autumn | 0.049 | 28.527 | 1.020 | 798.173 |
| | Summer | Contrast | 1 | . | . |
| | <1 | 0.050 | 0.442 | 0.195 | 0.998 |
| | 1–6 | 0.085 (N) | 1.923 | 0.913 | 4.050 |
| | >6 | Contrast | 1 | . | . |
| | | | | | |
| RSV | <1 | 0 | 9.477 | 4.968 | 18.081 |
| NBR | 1–6 | 0 | 11.382 | 5.989 | 21.630 |
| OT $p = 0.000$ | >6 | Contrast | 1 | . | . |
| | | | | | |
| HAdV | <1 | 0.009 | 0.411 | 0.211 | 0.801 |
| NBR | 1–6 | 0.014 | 2.211 | 1.178 | 4.152 |
| OT $p = 0.000$ | >6 | Contrast | 1 | . | . |
| | | | | | |
| HRV | <1 | 0.093 (N) | 1.698 | 0.915 | 3.152 |
| NBR | 1–6 | 0 | 3.242 | 1.771 | 5.934 |
| OT $p = 0.009$ | >6 | Contrast | 1 | . | . |
| | | | | | |
| MP | <1 | 0 | 0.131 | 0.067 | 0.258 |
| NBR | 1–6 | 0.436 (N) | 0.784 | 0.426 | 1.445 |
| OT $p = 0.000$ | >6 | Contrast | 1 | . | . |
| | Mean air temperature | 0.001 | 0.815 | 0.720 | 0.923 |

**Table 3** (*continued*)

| Dependent variables and statistical models | Independent variable | Significance | Exp(B) | 95% Wald confidence interval for Exp(B) | |
|---|---|---|---|---|---|
| | | | | **Lower limit** | **Upper limit** |
| SA | <1 | 0 | 4.858 | 2.383 | 9.908 |
| NBR | 1–6 | 0 | 4.393 | 2.129 | 9.063 |
| OT $p = 0.000$ | >6 | Contrast | 1 | . | . |
| | Average humidity | 0.017 | 0.871 | 0.777 | 0.975 |
| | Sunshine duration | 0.036 | 0.969 | 0.941 | 0.998 |
| | | | | | |
| KP | Spring | 0 | 0.059 | 0.015 | 0.233 |
| PR | Winter | 0.001 | 0.006 | 0 | 0.126 |
| OT $p = 0.000$ | Autumn | 0.001 | 0.035 | 0.005 | 0.245 |
| | Summer | Contrast | 1 | . | . |
| | Male | 0 | 2.104 | 1.492 | 2.967 |
| | Female | . | 1 | . | . |
| | <1 | 0 | 9.000 | 4.683 | 17.297 |
| | 1–6 | 0 | 4.900 | 2.482 | 9.673 |
| | >6 | Contrast | 1 | . | . |
| | Sunshine duration | 0 | 0.962 | 0.943 | 0.983 |
| | | | | | |
| HI | Male | 0.040 | 1.680 | 1.024 | 2.756 |
| NBR | Female | Contrast | 1 | . | . |
| OT $p = 0.000$ | <1 | 0.168(N) | 1.578 | 0.826 | 3.015 |
| | 1–6 | 0 | 3.199 | 1.726 | 5.932 |
| | >6 | Contrast | 1 | . | . |
| | | | | | |
| PA | Male | 0.002 | 2.692 | 1.424 | 5.089 |
| PR | Female | Contrast | 1 | . | . |
| OT $p = 0.000$ | <1 | 0.001 | 8.333 | 2.516 | 27.600 |
| | 1–6 | 0.002 | 6.667 | 1.981 | 22.435 |
| | >6 | Contrast | 1 | . | . |
| | | | | | |
| SP | <1 | 0.004 | 2.561 | 1.350 | 4.860 |
| NBR | 1–6 | 0 | 4.378 | 2.350 | 8.154 |
| OT $p = 0.001$ | >6 | Contrast | 1 | . | . |

## Lower respiratory pathogens

Table 6 showed the infection rates of various pathogens in children with LRTI. Analysis of infection rate of pathogens (Table 4) revealed that although RSV infection rates decreased with age, it remains the primary pathogen of LRTI in children under 6 years old. MP was the primary pathogen causing LRTI in children aged over six years, with its infection rate exceeding that of the overall infection rate. The infection rates of IFV-B and HAdV in LRTI were lower than the overall infection rate across all ages. The infection rate of bacterial pathogens in LRTI was slightly higher than the overall infection rate at all ages.

**Table 4  Table of distribution of respiratory pathogens by age and sex.**

| Pathogen | Age, gender, total (%) | | | | | |
|---|---|---|---|---|---|---|
| | <1 M 1223 | <1 F 722 | 1–6 M 2362 | 1–6 F 1641 | >6 M 963 | >6 F 759 |
| IFV-A | 46 (3.76%) | 26 (3.6%) | 201 (8.51%) | 131 (7.98%) | 97 (10.07%) | 83 (10.94%) |
| IFV-B | 41 (3.35) | 23 (3.18%) | 158 (6.69%) | 115 (7.01%) | 155 (16.10%) | 96 (12.65%) |
| RSV | 577 (47.18%) | 378 (52.35%) | 560 (23.71%) | 402 (24.50%) | 54 (5.61%) | 46 (6.06%) |
| HAdV | 55 (4.50%) | 30 (4.16%) | 305 (12.91%) | 197 (12.00%) | 214 (22.22%) | 164 (21.61%) |
| HRV | 191 (15.62%) | 99 (13.71%) | 356 (15.07%) | 236 (14.38%) | 100 (10.38%) | 74 (9.75%) |
| MP | 39 (3.19%) | 23 (3.19%) | 341 (14.44%) | 260 (15.84%) | 180 (18.69%) | 153 (20.16%) |
| | <1 M 463 | <1 F 260 | 1–6 M 548 | 1–6 F 396 | >6 M 179 | >6 F 154 |
| SA | 90 (19.44%) | 50 (19.23%) | 69 (12.59%) | 63 (15.91%) | 21 (11.73%) | 12 (7.79%) |
| KP | 56 (12.10%) | 29 (11.15%) | 31 (5.66%) | 18 (4.55%) | 8 (4.47%) | 2 (1.30%) |
| LP | 0 (0.00%) | 0 (0.00%) | 0 (0.00%) | 0 (0.00%) | 0 (0.00%) | 0 (0.00%) |
| HI | 173 (37.37%) | 82 (31.54%) | 342 (62.41%) | 231 (58.33%) | 122 (68.16%) | 99 (64.29%) |
| PA | 17 (3.67%) | 6 (2.31%) | 15 (2.74%) | 5 (1.26%) | 2 (1.12%) | 1 (0.65%) |
| SP | 282 (60.91%) | 150 (57.69%) | 449 (81.93%) | 326 (82.32%) | 126 (70.39%) | 104 (67.53%) |

With increasing age, the bacterial pathogen rate in LRTI declined, while the infection rates of IFV-B and HAdV gradually increased, and HRV gradually decreased. HI and SP data differed greatly from actual infection rates (refer to PCT results) and were listed here for reference.

# DISCUSSION

Statistical analysis of children attending the First People's Hospital of Tianshui City revealed that 35.8% were diagnosed with respiratory infections. Notably, the infection rate of COVID-19 remained low, aligning with prior studies, while the fatality rate of COVID-19 in children was exceptionally low (0.005%–0.01%) (*Howard-Jones et al., 2022*). Most respiratory pathogens exhibit seasonal peaks in winter. This phenomenon is attributable to seasonal dry and cold conditions and pathogens' biological traits. Data from Tables 3 and 4 showed that most pathogens exhibited age-related declines in infection rates, while positive rates of IFV-A, IFV-B, HAdV and MP specimens gradually increased with age. Among them, IFV-B and HAdV are typically restricted to upper respiratory tract infections, whereas RSV, MP and bacterial pathogens predominantly associate with lower respiratory tract infections. RSV and MP's capacity to induce lower respiratory tract infections may be related not only to their inherent pathogenicity but also the propensity for cause co-infection. The above phenomenon may be attributed to the developmental features of children's immune systems, physiological factors, and the pathogenicity of pathogens.

The Respiratory Tract Infection Etiology Surveillance Project Group of the Chinese Center for Disease Control and Prevention collected pathogen data of acute respiratory tract infection cases among preschool children (aged 1–6 years) from 2009 to 2019. The prevalence of pathogenic bacteria infection was RSV 25.7%, HRV 17.4%, IFV 14.2%,

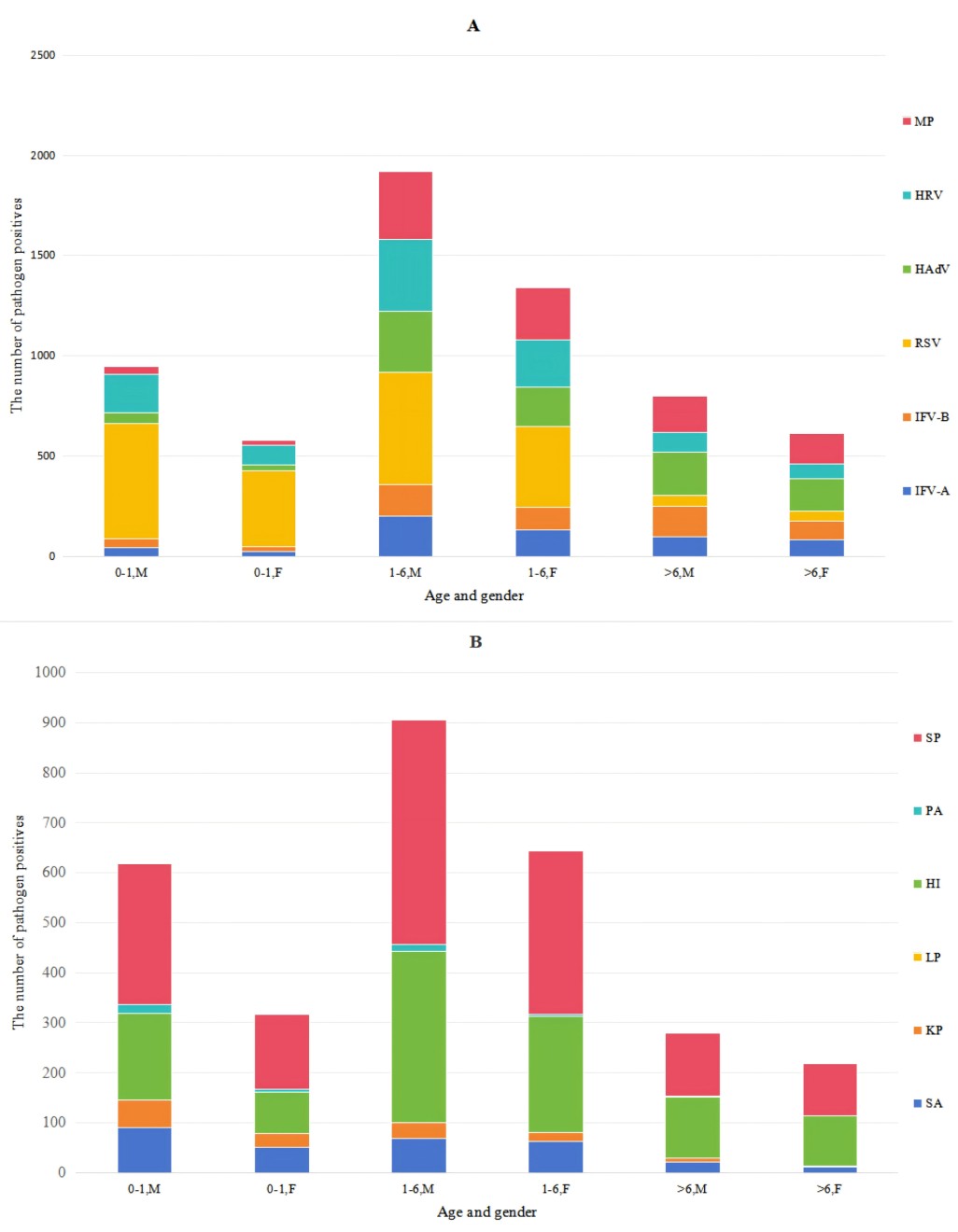

**Figure 2  Distribution of pathogens in boys and girls of different ages.** The pathogen color bars for each age group highly represent the number of infections.

HAdV 10.7%, MP 18.8%, SP 38.5%, HI 20.3%, SA 9.7%, and KP 7.4% (*Li et al., 2021*). The data of pathogenic bacterial infections in preschool children in this study were RSV 40.0%, HRV 18.4%, IFV 15.5%, HAdV 12.3%, MP 13.8%, SP 72.4%, HI 49.7%, SA 16.3%, and KP 8.0%. Compared to the National Center for Disease Control and Prevention's analysis, the infection rates of RSV, SP, HI and SA in this study were significantly higher than

| Pathogen | IFV-A | IFV-B | RSV | HAdV | HRV | MP | KP | PA | SA | HI | SP | LP |
|---|---|---|---|---|---|---|---|---|---|---|---|---|
| IFV-A | 58 | | | | | | | | | | | |
| IFV-B | 2 | 41 | | | | | | | | | | |
| RSV | 7 | 1 | 280 | | | | | | | | | |
| HAdV | 5 | 2 | 10 | 88 | | | | | | | | |
| HRV | 6 | 1 | 28 | 6 | 113 | | | | | | | |
| MP | 11 | 4 | 11 | 12 | 8 | 142 | | | | | | |
| KP | 3 | 0 | 21 | 4 | 9 | 7 | 76 | | | | | |
| PA | 2 | 1 | 8 | 0 | 2 | 1 | 9 | 31 | | | | |
| SA | 5 | 1 | 45 | 5 | 21 | 12 | 19 | 8 | 148 | | | |
| HI | 40 | 31 | 112 | 72 | 66 | 87 | 30 | 11 | 57 | 519 | | |
| SP | 47 | 33 | 197 | 71 | 80 | 107 | 63 | 23 | 123 | 413 | 705 | |
| LP | 0 | 0 | 0 | 0 | 0 | 0 | 0 | 0 | 0 | 0 | 0 | 0 |

**Figure 3 Number of co-infected cases.** In order to analyze the relationship between the two pathogens, the pathogens in the cases infected by more than two pathogens were pairwise separated. The data in the figure were the number of pathogen positives.

those in the national data, whereas MP's infection rate slightly decreased. During 2011, the Gansu Provincial Center for Disease Control and Prevention collected respiratory tract samples from 279 children under 12 years old in Lanzhou City (approximately 400 kilometers from Tianshui City). The positive rates for RSV, HRV, HAdV, and IFV-A were 12.9%, 10.4%, 4.7%, and 2.2%, respectively (*Huang et al., 2013*). In our study, RSV, HRV, HAdV, and IFV-A were detected in this population with prevalence rates of 26.3%, 13.7%, 12.5%, and 7.6%, respectively. The comparative data demonstrate that the positive rates of these four pathogens in this study were significantly higher than those in data of Lanzhou. A comparative analysis of comprehensive data demonstrates that the infection rate of respiratory pathogens among children significantly increased post-outbreak of novel coronavirus pneumonia. The primary factor behind the above phenomenon is prolonged home confinement during the COVID-19 pandemic, which decreased exposure risk to respiratory pathogens. Pathogen contact stimulation plays a critical role in immune system development in children and adaptive immunity establishment. Other influencing factors include reduced physical activity duration, decreased sunlight exposure, and depressed mood, *etc*.

The aforementioned infection characteristics and epidemic patterns of respiratory pathogens are likely attributable to the following biological mechanisms. The dry and cold

Table 5 **Statistical analysis table of pathogen co-infection data.** 950 children (aged <13 years) with 12 kinds of pathogens were detected at the same time, and 221 (23.3%) were co-infected. Binary LOGISTCI regression model was used for analysis, backward LR, HL test $P > 0.05$ were used as inclusion criteria.

| Dependent variable and HL | Independent variable | Significance | Exp(B) | 95% Wald confidence interval for Exp(B) | |
|---|---|---|---|---|---|
| | | | | Lower limit | Upper limit |
| IFV-A | RSV | 0.010 | 2.906 | 1.292 | 6.536 |
| HL = 0.536 | HI | 0.086 | 0.602 | 0.337 | 1.074 |
| | | | | | |
| IFV-B | RSV | 0.003 | 20.651 | 2.797 | 152.468 |
| HL = 0.857 | HRV | 0.047 | 7.655 | 1.029 | 56.945 |
| | MP | 0.037 | 3.093 | 1.071 | 8.934 |
| | SA | 0.047 | 7.620 | 1.025 | 56.644 |
| | | | | | |
| RSV | HAdV | 0.000 | 3.581 | 1.789 | 7.169 |
| HL = 0.960 | HRV | 0.043 | 1.623 | 1.015 | 2.596 |
| | MP | 0.000 | 6.771 | 3.559 | 12.881 |
| | HI | 0.000 | 1.993 | 1.472 | 2.699 |
| | SA | 0.089 | 1.420 | 0.948 | 2.127 |
| | IFV-A | 0.004 | 3.358 | 1.469 | 7.676 |
| | IFV-B | 0.003 | 21.525 | 2.917 | 158.839 |
| | | | | | |
| HAdV | HRV | 0.042 | 2.473 | 1.033 | 5.924 |
| HL = 0.823 | HI | 0.000 | 0.287 | 0.163 | 0.508 |
| | SA | 0.028 | 2.864 | 1.120 | 7.325 |
| | RSV | 0.000 | 3.461 | 1.731 | 6.920 |
| | | | | | |
| HRV | MP | 0.005 | 2.941 | 1.387 | 6.238 |
| HL = 0.990 | RSV | 0.03 | 1.667 | 1.051 | 2.646 |
| | | | | | |
| MP | SA | 0.009 | 2.325 | 1.233 | 4.386 |
| HL = 0.413 | IFV-B | 0.050 | 2.884 | 1.002 | 8.300 |
| | RSV | 0.000 | 6.824 | 3.603 | 12.923 |
| | HRV | 0.005 | 2.965 | 1.395 | 6.302 |
| | | | | | |
| KP | SP | 0.035 | 0.509 | 0.271 | 0.953 |
| HL = 0.867 | HI | 0.011 | 1.897 | 1.161 | 3.098 |
| | PA | 0.000 | 0.217 | 0.094 | 0.500 |
| | | | | | |
| SP | HI | 0.000 | 0.486 | 0.360 | 0.657 |
| HL = 0.937 | SA | 0.002 | 0.475 | 0.298 | 0.758 |
| | KP | 0.048 | 0.531 | 0.284 | 0.993 |

| Dependent variable and HL | Independent variable | Significance | Exp(B) | 95% Wald confidence interval for Exp(B) | |
|---|---|---|---|---|---|
| | | | | Lower limit | Upper limit |
| HI | SA | 0.000 | 2.129 | 1.462 | 3.101 |
| HL = 0.895 | RSV | 0.000 | 2.088 | 1.550 | 2.811 |
| | HAdV | 0.000 | 0.302 | 0.170 | 0.535 |
| | KP | 0.011 | 1.915 | 1.159 | 3.162 |
| | SP | 0.000 | 0.516 | 0.379 | 0.703 |
| PA | KP | 0 | 0.213 | 0.094 | 0.484 |
| HL = 0.997 | | | | | |
| SA | IFV-B | 0.045 | 7.781 | 1.051 | 57.581 |
| HL = 0.929 | HAdV | 0.026 | 2.893 | 1.137 | 7.361 |
| | MP | 0.014 | 2.19 | 1.169 | 4.102 |
| | SP | 0.001 | 0.45 | 0.282 | 0.719 |
| | HI | 0 | 2.092 | 1.443 | 3.034 |

**Table 6  Pathogen data in children with lower respiratory infections of different ages.**

| Pathogen | Age and total (%) | | |
|---|---|---|---|
| | <1 360 | 1–6 257 | >6 139 |
| IFV-A | 30 (8.3%) | 26 (10.1%) | 15 (10.8%) |
| IFV-B | 10 (2.8%) | 9 (3.5%) | 12 (8.6%) |
| RSV | 227 (63.1%) | 112 (43.6%) | 26 (18.7%) |
| HAdV | 28 (7.8%) | 20 (7.8%) | 26 (18.7%) |
| HRV | 81 (22.5%) | 46 (17.9%) | 21 (15.1%) |
| MP | 17 (4.7%) | 33 (12.8%) | 88 (63.3%) |
| SA | 64 (17.8%) | 44 (17.1%) | 16 (11.5%) |
| KP | 50 (13.9%) | 17 (6.6%) | 8 (5.8%) |
| LP | 1 (0.3%) | 0 (0.0%) | 0 (0.0%) |
| HI | 149 (41.4%) | 152 (59.1%) | 90 (64.7%) |
| PA | 17 (4.7%) | 6 (2.3%) | 3 (2.2%) |
| SP | 232 (64.4%) | 218 (84.8%) | 98 (70.5%) |

climatic conditions contribute to enhanced survival capacity and transmission potential of respiratory pathogens. Meanwhile, such extreme climatic conditions exert significant detrimental effects on the respiratory mucosal barrier, thereby compromising its physical defense and immune surveillance functions and facilitating pathogen invasion through the physiological barrier (*Mettelman, Allen & Thomas, 2022*). Newborns possess maternal antibodies with varying concentrations and effects against different pathogens, whose levels progressively decline with age, thereby reducing early passive immune protection ability. The lymphatic system of children aged 1–6 remains in a developmental phase, with their humoral and cellular immunity functions still immature (*Chapman & Chi, 2022*; *Pieren, Boer & De Wit, 2022*). As children age, the expected incidence of infections

gradually decreases. Beyond the immune system as a key factor, the progressive maturation of their respiratory tracts and skin structure and function, along with evolving dietary habits, may also contribute to this phenomenon. The mutual inhibition or promotion of co-infection among respiratory pathogens is primarily governed by their biological characteristics and ecological niches. Competitive inhibition necessarily occurs among pathogens sharing the same ecological niche. Interactions among pathogens occupying distinct ecological niches may synergistically enhance their pathogenicity and survival capabilities. This constitutes a key factor underlying respiratory pediatric associated with respiratory pathogen co-infections, mediated through mechanisms involving mucosal barrier, immune dysregulation, and compromised host defense mechanisms (*Blyth et al., 2013*; *Rowe & Rosch, 2021*).

Through analysis of pre- and post-pandemic pediatric respiratory pathogen surveillance data, this study establishes an evidence base for public health authorities to monitor transmission dynamics, assess systemic impacts, and inform strategic responses. Epidemiological data on respiratory pathogens provide clinicians with diagnostic guidance for optimized treatment strategies, while supplying critical evidence for investigating transmission dynamics, infection pathways, infection pathways, pathogenic mechanisms, and preventive intervention development.

However, this study's data was derived predominantly from individuals with clinically significant respiratory infections, excluding mild or latent infections, which may introduce potential population bias. Population mobility constitutes a key determinant of respiratory pathogen infection, amplified by China's recent tourism expansion. Furthermore, epidemiological surveillance data showing pathogen detection rates require the actual trend of cautious interpretation regarding actual infection trends. Interpretation should account for testing population size, particularly given the marked rise in respiratory infection cases in recent years. Despite potential statistical biases, a marked shift persists in pediatric respiratory pathogen acquisition rates pre- and post-COVID-19 pandemic. Prolonged pathogen exposure deprivation has altered epidemiological patterns of pediatric respiratory pathogen acquisition. In addition to the lag in children's immune systems and development and their ability to respond to corresponding pathogens caused by long-term isolation from pathogen exposure, are these differences related to immune system disorders, multiple organ damage and corresponding clinical consequences caused by COVID-19 infection? These findings necessitate further clinical validation.

## CONCLUSIONS

This study analyzed respiratory pathogen infection data in children pre- and post-pandemic to assess pathogenicity, epidemiological trends and epidemiological influencing factors, determining that prolonged pathogen isolation increases children's susceptibility to respiratory pathogens. At the same time, this study also provided data to support for public health department formulating scientific and effective prevention strategies and related research.

## Funding

This work was supported by Natural Science Foundation of Gansu Province (No. 24JRRE013). The funders had no role in study design, data collection and analysis, decision to publish, or preparation of the manuscript.

## Grant Disclosures

The following grant information was disclosed by the authors:
Natural Science Foundation of Gansu Province: 24JRRE013.

## Competing Interests

The authors declare there are no competing interests.

## Author Contributions

- Ping Chen conceived and designed the experiments, analyzed the data, authored or reviewed drafts of the article, and approved the final draft.
- Yahong Li performed the experiments, authored or reviewed drafts of the article, and approved the final draft.
- Shiwei Li conceived and designed the experiments, analyzed the data, prepared figures and/or tables, authored or reviewed drafts of the article, and approved the final draft.
- Xin Hua performed the experiments, authored or reviewed drafts of the article, and approved the final draft.
- Yu Liu performed the experiments, analyzed the data, prepared figures and/or tables, authored or reviewed drafts of the article, and approved the final draft.

## Human Ethics

The following information was supplied relating to ethical approvals (i.e., approving body and any reference numbers):

The Ethics Committee of Tianshui First People's Hospital approved the study (2023-07).

## Data Availability

The raw measurements are available in the uploaded tables and graphs.

## Supplemental Information

Supplemental information for this article can be found online at http://dx.doi.org/10.7717/peerj.19710#supplemental-information.

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
