# Peer review of "Analysis of epidemic trend of respiratory pathogens in children after long-term pathogen isolation"

_PeerJ, doi:10.7717/peerj.19710_

## Round 0.1 · original submission · Major Revisions

Please address the comments of all reviewers and provide point-wise responses along with the revised manuscript.

**Language Note:** The review process has identified that the English language must be improved. PeerJ can provide language editing services - please contact us at [email protected] for pricing (be sure to provide your manuscript number and title). Alternatively, you should make your own arrangements to improve the language quality and provide details in your response letter. – PeerJ Staff

·

Basic reporting

1, The submission does not comply with PeerJ’s guidelines.
https://peerj.com/about/author-instructions/#instruction-standard-sections-rass
・The Abstract does not adhere to the requirements outlined in the guidance.
・In the manuscript, the funding information should not be used to acknowledge funders – funding will be entered online in the declarations page as a separate Funding Statement and appear on the published paper.
・There are minor grammar mistakes.
ex. no space, first of sentence is large or small letter, etc.

Recommendation: The authors used PeerJ manuscript guidelines and a professional language editing service may help refine the manuscript further for international audiences.

2, Literature References and Context
The manuscript provides sufficient references to support the study's background. Citations from reputable sources (e.g., the World Health Organization and systematic reviews in The Lancet) are appropriate and relevant.
Strengths: The introduction effectively contextualizes the significance of respiratory infections in children and the impact of COVID-19-related restrictions.
Suggestion: • Expand on the knowledge gap this study aims to fill, specifically regarding the long-term effects of isolation on immune responses in children (lines 46-47).

3, Article Structure
The structure adheres to PeerJ standards, with clear sections: Abstract, Introduction, Methods, Results, Discussion and Conclusion.
・In the manuscript, these figure and table legends should be deleted.
ex)
L. 85 Table 1. The infection of pathogenic bacteria changed with time.
L. 87 Figure 1. Weather data of Tianshui.
L. 89 Table 2. Relationship between pathogen infection rate and environmental factors. * <0.05
** <0.01 0 No correlation
Relationship between pathogens and environmental factors

4, Self-Contained and Relevant Results
The results are directly tied to the hypotheses and provide comprehensive insights into pathogen trends. However, a more straightforward link between findings and their broader implications (e.g., for public health policies) would strengthen the study with references or guidelines in the authors' country.

5, Raw Data Sharing
The inclusion of raw data aligns with journal guidelines. Ensure the dataset metadata includes comprehensive descriptions for replication purposes.

Experimental design

1. Scope and Research Question
While the manuscript addresses an important topic within the journal's aims and scope, the research question is not clearly stated in the manuscript. The authors should explicitly define the research question and identify the specific knowledge gap their study seeks to fill. This will help contextualize the study's importance and contribution to the field.

2. Rigor and Ethical Standards
・The manuscript states that the study was conducted under ethical approval (IRB approval number 2023-07). However, further details on obtaining informed consent should be provided to ensure transparency. These ethical statements should be replaced after the Study population in Materials & methods.
・Data integrity and quality control measures during collection and analysis would be beneficial.
・I can't clearly understand what the test in blood cell analysis and related examination indicated ARIS, and there seems to be no indication.

3. Reproducibility
The methods are described in detail, which generally allows for reproducibility. However, certain areas could benefit from additional information:
・Specify whether seasonal and environmental factors were controlled or adjusted statistically during analysis.
・Describe any steps taken to minimize bias during data collection and pathogen detection.
・Ensure that the laboratory protocols, including the calibration and validation of instruments, are thoroughly explained.

Validity of the findings

1. Clarity in Results Presentation
・MIQE specification files (Table 6) were essential to link the method section in manuscript.
・The correlation analysis (e.g., Table 2) requires further elaboration, particularly on the strength and direction of the correlations.
・Graphical representations of seasonal trends (e.g., infection rates by age or pathogen type) would improve clarity and accessibility compared to tables alone.

2. Statistical Reporting
Tables and figures are needed raw data (recommended supplemental files) and clear p-value.
・In Figure 1, ensure all text in annotations is legible and professional. Please ensure that the significant figures in Figure 1 are appropriate and consistent with scientific standards. Adjustments may be needed to enhance the accuracy and clarity of the data presentation.
・The all figure data is difficult to read, and converting it into two-dimensional graphs would improve clarity. It would be helpful to include the corresponding data tables beneath the graphs.
・Correlation (R) was not found.
・P-values should be reported to the fourth decimal place. This level of precision helps assess the statistical significance more accurately.

3. Context and Relevance
・While the findings are interesting, the discussion lacks depth regarding how they align with or diverge from previous studies. Expanding this section would better validate the study’s contributions to the field.
・Further exploration of potential confounding variables, such as socioeconomic factors or regional healthcare differences, would strengthen the robustness of the findings.

4. I don't find Conclusions.

Additional comments

The submission does not fully comply with PeerJ’s guidelines.
https://peerj.com/about/author-instructions/#instruction-standard-sections-rass

I sincerely regret to point out that the manuscript does not adhere to the formatting requirements of PeerJ's submission guidelines. Consequently, it is possible that this review may not have addressed all issues related to the content.

Reviewer 2 ·

Basic reporting

The manuscript aims to analyze the trends of respiratory infections in children following long-term pathogen isolation. It examines important factors such as age, seasonal variation, and co-infections to better understand how these infections spread and inform public health measures. The topic is highly relevant, especially considering the potential long-term effects of COVID-19-related isolation on respiratory health in children.

While the abstract presents the key findings, the structure could be improved to clearly separate age trends, seasonal patterns, and co-infections for better readability. Additionally, the term "long-term pathogen isolation" requires clarification. Is it referring to social distancing, reduced exposure to pathogens, or other factors? A stronger emphasis on the study’s public health significance and the implications of the findings would also be beneficial.

In the introduction, the background on the global burden of ARIs and the impact of COVID-19 measures on respiratory infections in children is well established. However, the section would benefit from a clearer flow, including a more precise research gap and more focused objectives. The phrase "providing effective data" should be replaced with specific examples of the data being provided, such as age-specific infection patterns or seasonal trends.

In terms of figures and tables, significant improvements are needed. There are mismatches in figure numbering (Figure 3 as 2b, etc), inconsistent fonts, and corrupted text. Figure 4 is particularly blurry, which makes interpretation difficult. A careful revision of this section is required to ensure clarity, consistency, and proper data visualization.

Experimental design

Regarding methodology, the study provides an overview but lacks sufficient details needed for clarity and reproducibility. Specific diagnostic techniques for pathogen detection and inclusion criteria for ARIs are not clearly outlined. The use of only Pearson’s χ² test for analysis is inadequate, as it does not address the complexity of seasonal variations or co-infections. The study mentions environmental variables but does not specify how these were incorporated into the analysis, such as through regression models or correlation tests.

Validity of the findings

The results section lacks statistical rigor and clarity. Figures are not cross-referenced, making the interpretation difficult. Stronger analyses, such as regression models for environmental factors and significance testing for co-infections, are needed. Explicitly stating key findings rather than relying solely on figures would improve clarity.

The discussion needs to be restructured, with stronger statistical support and more precise analysis. Currently, it lacks depth and makes unverified claims. The section on co-infections and seasonal trends is underdeveloped and lacks mechanistic insights. The impact of COVID-19 is mentioned but not critically analyzed. The final paragraph, which contains details on methodology, should be moved to the methods section.

Additional comments

The supplemental data is not cited anywhere in the manuscript, making its inclusion unclear. If it contains relevant information, it should be referenced appropriately within the main text. Otherwise, it's presence is unnecessary.

Lastly, minor formatting issues persist. The text should be justified for a cleaner presentation. Additionally, spacing issues, such as with "thenational" on line 27, and typographical errors like "ARIS" instead of "ARIs" on line 62, need correction.

In conclusion, the manuscript presents an interesting and relevant study. However, it requires significant revision to improve clarity, scientific rigor, and structure.

---

## Round 0.2 · Major Revisions

Please address all comments made by both reviewers. Comments made by Reviewer 1 during 1st review have also not been fully addressed. Please make sure all comments are properly addressed and submission is following journal guidelines before submitting revised manuscript.

·

Basic reporting

See the below comment.

Experimental design

See the below comment.

Validity of the findings

See the below comment.

Additional comments

I sincerely apologize, but I kindly request that you resubmit the fully revised manuscript.

Reviewer 2 ·

Basic reporting

I appreciate the author's efforts to revise the manuscript in response to prior feedback. Several improvements have been made, including updates to the statistical analysis, clearer definition of ARIs, and more detailed methodology regarding sample processing and environmental variables. The use of Poisson, negative binomial, and logistic regression is more appropriate for the data structure, and figure/table presentation has been improved.
However, significant issues remain that continue to affect the manuscript's clarity and scientific rigor:
1. Language and Structure: Despite minor edits, the manuscript still requires extensive language polishing. Grammatical errors, awkward phrasing, and inconsistent formatting are present throughout. The Results and Discussion sections remain poorly organized and overly verbose, often burying key findings in unclear or redundant text.

2. Results Section: While statistical methods have been updated, the narrative remains hard to follow. Tables and figures (also sub-panels A, B) are not always clearly introduced or interpreted in-text. Paragraphs are lengthy and often combine multiple unrelated findings, making the section dense and difficult to navigate.

3. Discussion Section: This section lacks a clear structure and includes speculative content not supported by data (e.g., anecdotal claims about post-COVID immune dysfunction). Recommendations for revision: Reorganize the Discussion into clear subsections, such as: Interpretation of key findings, Comparison with prior studies, Biological mechanisms (brief), Public health implications, Study limitations
• Remove speculative or anecdotal content that lacks evidence.
• Condense background discussion to what directly supports the data.
• Cite more recent and relevant literature.
• Carefully proofread the section to resolve grammatical and stylistic issues.

4. Conclusion: The revised conclusion is still vague. It should directly summarize the most important findings and their relevance to pediatric public health without generalizations.

5. References: The manuscript lists all authors for references, even when there are dozens or more, leading to entries that span multiple pages (e.g., lines 362–374, 377–394, and 394–511). This approach is inconsistent with PeerJ's guidelines and standard academic practices. This oversight significantly undermines the manuscript's professionalism and must be corrected.

6. The manuscript includes supplementary materials that provide additional data supporting the study's findings. However, these materials are not adequately cited within the main text. According to PeerJ's guidelines, all supplementary files should be appropriately referenced in the manuscript to ensure transparency and ease of access for readers. For example, supplementary figures should be cited as "Fig. S1," tables as "Table S1," and datasets as "Data S1" at relevant points in the Methods and Results sections.

Recommendation: Major Revision
The study addresses an important question with valuable data, but the manuscript still requires substantial restructuring and professional editing to meet publication standards. I encourage the authors to revise with attention to clarity, conciseness, and alignment with scientific writing norms.

Experimental design

NA

Validity of the findings

NA

Additional comments

NA

---

## Round 0.3 · accepted · Accept

The authors have addressed most of the reviewers' comments and manuscript can be published after proofreading.

·

Basic reporting

none

Experimental design

none

Validity of the findings

none

Additional comments

none